# Music Listening, Emotion, and Cognition in Older Adults

**DOI:** 10.3390/brainsci12111567

**Published:** 2022-11-17

**Authors:** Margherita Vincenzi, Erika Borella, Enrico Sella, César F. Lima, Rossana De Beni, E. Glenn Schellenberg

**Affiliations:** 1Department of General Psychology, University of Padova, 35131 Padova, Italy; 2Centro de Investigação e Intervenção Social (CIS-IUL), Instituto Universitário de Lisboa (ISCTE-IUL), 1649-026 Lisboa, Portugal; 3Institute of Cognitive Neuroscience, University College London, London WC1N 3AZ, UK; 4Department of Psychology, University of Toronto Mississauga, Mississauga, ON L5L 1C6, Canada

**Keywords:** music, working memory, aging, cognition, emotion, listening, executive function

## Abstract

Using the arousal and mood hypothesis as a theoretical framework, we examined whether community-dwelling older adults (*N* = 132) exhibited cognitive benefits after listening to music. Participants listened to shorter (≈2.5 min) or longer (≈8 min) excerpts from recordings of happy- or sad-sounding music or from a spoken-word recording. Before and after listening, they completed tasks measuring visuospatial working memory (WM), cognitive flexibility and speed, verbal fluency, and mathematical ability, as well as measures of arousal and mood. In general, older adults improved from pre- to post-test on the cognitive tasks. For the test of WM, the increase was greater for participants who heard happy-sounding music compared to those in the other two groups. The happy-sounding group also exhibited larger increases in arousal and mood, although improvements in mood were evident only for the long-duration condition. At the individual level, however, improvements in WM were *unrelated* to changes in arousal or mood. In short, the results were partially consistent with the arousal and mood hypothesis. For older adults, listening to happy-sounding music may optimize arousal levels and mood, and improve performance on some cognitive tasks (i.e., WM), even though there is no direct link between changes in arousal/mood and changes in WM.

## 1. Introduction

Age-related declines in cognitive abilities pose a threat to the wellbeing and quality of life of older adults, such that interventions with even a modicum of success are important to document. Studies on aging have, in fact, focused on developing interventions that promote active aging and/or counteract, or at least delay, age-related cognitive decline, devoting particular attention to abilities that decline with age (e.g., working memory, executive functions) [1,2]. Thus, it is important to identify factors that could influence and sustain cognitive performance in older adults in order to promote autonomy and quality of life in older age. In the present investigation, we tested a sample of community-dwelling older adults, asking whether listening to brief excerpts of happy or sad music had positive effects on emotional states and cognitive abilities.

### 1.1. Background Music

Effects of music listening on cognition and emotion have been studied when music is played either (1) *while* participants complete a task (i.e., background music) or (2) *before* they start the task. Research on background music focuses on whether it affects performance in the concurrent primary task, such as reading comprehension or driving ability. Results are equivocal, presumably because of large individual differences (e.g., personality) and contextual factors (e.g., task difficulty) [3,4]. Indeed, meta-analyses reveal null effects when different primary tasks are analyzed together [5]. When examined separately, however, background music appears to enhance achievements in sports and improve affect, although it has small detrimental effects on reading comprehension and memory [5].

Older adults have WM and executive-function deficits [6] that have been attributed to an impairment that occurs during aging. One hypothesis—that older adults have difficulty inhibiting irrelevant information during encoding and retrieval [7,8]—has equivocal empirical support [9,10,11]. In any event, when older adults are asked to perform classic WM tasks, such as the reading-span test [12], they produce more memory-intrusion errors compared to young adults by recalling non-target words [13]. Thus, background music could have a stronger negative impact on older compared to younger adults because it is more difficult for older adults to ignore. Indeed, older, but not younger, adults’ associative memory is impaired in the presence of background music, even though both age groups self-report that the music is distracting [14]. One possibility is that positive results among older adults are evident only with relatively low-amplitude background music. For younger adults, loud background music is more disruptive to reading comprehension than the same music presented at a lower volume [15].

For these reasons, in the present study, we did *not* present music concurrently with the cognitive tasks that we administered to our sample of older adults. Rather, participants completed the tasks *after* they listened to music. They also completed parallel versions of the same tasks before the music in order to obtain a baseline measure of performance, and so that we could monitor changes in cognitive performance as a function of music listening.

### 1.2. Cognitive Performance after Music Listening

The study of cognitive performance after listening to music stems from two separate research traditions that eventually merged. One is based on mood induction [16] and the other on the so-called “Mozart effect” [17]. The mood-induction literature confirms that listening to music influences self-reports and physiological indices of mood, which, in turn, influence consumer behavior, as well as performance on IQ subtests and other cognitive tasks (e.g., writing speed, decision making) [16]. Although mood-inducing film clips induce emotions differentially for younger and older adults [18], emotional responses to music tend to be more intense among older compared to younger adults for happy-sounding music (e.g., Abba’s *Dancing Queen*) but slightly weaker for sad-sounding music (e.g., Billie Holiday’s *Strange Fruit*) [19]. Regardless, emotional responses to music are evident in older adulthood. In fact, after negative mood is induced, self-selected music has a stronger ameliorating effect for older than for younger adults [20], although the *perception* of emotions conveyed by music may be skewed positively among older adults [21,22].

In contrast, original studies of the Mozart effect [23,24] focused solely on cognitive outcomes, ignoring the mediating factor of emotional state, assuming that listening to classical music directly activates brain areas that enhance cognition (for a review, see [17]). The effect was attributed to Mozart specifically following the 1993 publication of a newsworthy report: performance on spatial tasks was enhanced after listening to music composed by Mozart compared to control conditions that involved listening to relaxation instructions or sitting in silence [23]. The authors’ interpretation was one of priming effects between domains that have no obvious connection (music, spatial ability). Subsequent research revealed that the effect is indeed mediated by emotional state—specifically, arousal and mood [25]—which can be influenced by other types of music (e.g., play-songs for young children, pop-songs for older children) and nonmusical stimuli (e.g., a narrated story) [26,27,28]. It also extends to performance on non-spatial tasks (e.g., processing speed, creativity) [28,29]. The specific Mozart piece used in the original and subsequent studies is composed in a major mode, which improves mood, with a fast tempo, which increases arousal [30]. The well-known circumplex model of emotions posits that all emotions can be mapped onto a two-dimensional (arousal and mood/valence) space [31]. The *arousal and mood hypothesis* [25] extends this view, proposing that music often causes changes in arousal and/or mood, which in turn influence cognitive performance. In other words, the link between music and cognition is proposed to be mediated by emotion.

### 1.3. The Present Study

Our main aim was to examine whether community-dwelling, typically aging older adults show emotional and cognitive benefits after listening to music. The arousal and mood hypothesis provided our theoretical framework. To the best of our knowledge, ours is the first study to test this hypothesis in a sample of older adults. One previous study claimed to do so but there was no actual measure of emotional response, and the music was presented concurrently with the cognitive tasks, which meant that attentional limitations could have affected cognitive performance [32]. Our musical stimuli were the same as those used previously [23,25,29,33]—not because they are particularly special in a musical sense but because they are instrumental, clearly happy- or sad-sounding, and have been used successfully to affect arousal and/or mood and, subsequently, cognition. For the control condition, following Nantais and Schellenberg [26], we chose a spoken-word recording to compare music listening with an auditory stimulus that was similar in terms of engagement and attentional capture. In addition to measuring arousal and mood before and after music listening, we also measured dominance—the extent to which the sensations and affective reactions evoked by a stimulus are controllable [34]. Dominance is considered by some theorists to be a third dimension of emotional responding [35,36]. We hypothesized that the music manipulations would affect arousal and mood but not dominance, thereby providing evidence of discriminant validity for the arousal and mood hypothesis.

Our dependent cognitive measures were tests of WM and executive functions (verbal fluency, cognitive flexibility, and speed), and a test of arithmetic ability. We focused on WM and executive functions because of the declines observed in older individuals, which extend to healthy aging, mild cognitive impairment, and Alzheimer’s disease [6,37,38,39]. WM, in particular, declines linearly and steadily across the adult lifespan [40]. Thus, manipulations that have positive effects for older adults might also be informative for the development of interventions for sustaining active aging and limiting age-related cognitive changes. We also included a test of arithmetic abilities to determine whether any observed effects might extend to abilities that are required in everyday life. Deficits in simple arithmetical abilities are evident in both normal and pathological aging [41], such that it is particularly important to identify factors that could have positive effects.

We also asked an exploratory question: whether listening for a smaller amount of time (e.g., 2.5 min) would have the same impact as longer-duration listening (e.g., 8.2 min) on cognition and emotion. To date, listening duration has been neglected as a contributing factor in studies of music listening and cognitive performance. Indeed, in previous research, duration varied from a few seconds [42] to 10 min [43], even though, at some point (e.g., milliseconds vs. one hour), duration must play a role. Indeed, when individuals listen to music to reduce stress, benefits are evident only when the listening is at least 20 min per day [44]. Shorter-term effects of music listening might be particularly time-sensitive for older adults, who generally need more time to focus and process auditory stimuli [45]. In any event, regardless of the listeners’ age, the findings could inform the designs of future laboratory studies of music listening, for which time constraints almost always play a role.

In line with the arousal and mood hypothesis [25], we expected to find a positive effect of listening to music on arousal and mood (but not dominance), which could, in turn, lead to improvements in cognitive performance. In particular, we expected to find enhancements in arousal and mood after listening to a happy-sounding music excerpt but the opposite effects for sad-sounding music, and no changes in the control condition. We also expected enhanced cognitive performance only after listening to the happy-sounding music. We had no predictions for the duration manipulation, which was exploratory. As older people generally need more time to process auditory stimuli [45], however, our sample could benefit from longer compared to shorter exposure to music.

## 2. Materials and Methods

### 2.1. Participants

A total of 132 older-adult volunteers (61 female) between 65 and 75 years (*M* = 69.39, *SD* = 3.06) met our inclusion criteria. All were healthy, native Italian-speaking, community-dwelling individuals, recruited through word of mouth or from associations for older adults in northeast and southern Italy. None was a professional musician, and all reported listening to classical music only occasionally. Education varied from 5 to 20 years (*M* = 10.56, *SD* = 4.01). A power analysis confirmed that an estimated sample of 18–20 in each group provide power of 0.80 for an effect size of 0.30 and a *p* value less than 0.05 using the R 4.2.2 software’s *pwr* library (R Foundation for Statistical Computing, Vienna, Austria, 2021).

Participants were required to have good physical and mental health as assessed with a semi-structured interview (including questions regarding autonomy), as well as good cognitive functioning, using a cutoff score of 8 on the Italian Checklist for Multidimensional Assessment (SVAMA). The SVAMA provided a multidimensional evaluation (including questions regarding memory and time/place orientation) [46]. No participant scored significantly below age- and education-matched norms in the Vocabulary subtest from the Wechsler Adult Intelligence Scale—4th Edition. Pre-screening also included a psychological wellbeing measure (Psychological Wellbeing Questionnaire (PWB-Q), personal satisfaction subscale) [47]. All participants scored above the established threshold for low personal satisfaction. We also used the Positive and Negative Affect Scale (PANAS) [48] to measured trait-positive and -negative emotional states. Finally, the Adaptive Functions of Music Listening Scale (AFML) [49] was administered to confirm that music-listening habits did not differ across groups. Scores on the AFML represent the degree to which music listening is used adaptively (e.g., for stress regulation, to evoke emotional experiences).

### 2.2. Auditory Stimuli

Two pieces of music—the same as those used in previous research [25,29,33,50]—were excerpted from recordings of classical music. One was the first movement (*Allegro con spirito*) from Mozart’s sonata K 448 for two pianos in D major, performed by Radu Lupu and Murray Perahia. This piece is in a major mode with a relatively fast tempo. Excerpts with these musical characteristics have been shown to induce higher levels of arousal, improve mood, and enhance cognitive performance [25,30,33,51]. The other piece was Albinoni’s *Adagio in G minor* for organ and strings, performed by I. Solisti Veneti and conducted by Claudio Scimone. (Although the *Adagio* is commonly attributed to the Baroque composer Tomaso Albinoni, it is now thought to have been composed in the mid-20th century by the Italian musicologist Remo Giazotto). It is in a minor mode with a slow tempo. Excerpts with these characteristics can lower arousal levels, decrease positive mood (or increase negative mood), and lead to small declines (or no change) in cognitive performance [25,28,33,50]. Such effects can emerge simply by manipulating the tempo and mode of a single piece [30], and they are not limited to our musical stimuli [52]. Moreover, they represent general trends, not hard-and-fast rules. For example, many pieces composed in a minor mode do not sound sad (e.g., Kylie Minogue’s *Can’t Get You Out of My Head*).

We used Audacity 3.2.1 software (The Audacity Team, Pittsburgh, PA, USA, 2022) to create two versions of both excerpts: a longer one (≈8.2 min) and a shorter one (≈2.5 min). The excerpts were transferred digitally from CD without loss of sound quality (44.1 kHz, 16 bit). For the Mozart sonata, the longer version comprised the first movement minus the first refrain (repetition) so that it would not end abruptly and would be approximately the same duration (8:31) as the longer version of the Albinoni *Adagio* (8:16), for which the piece was played in full. For the shorter versions, we used the first refrain of the Mozart sonata K 448 (2:26) and two repetitions of the *Adagio* main theme (2:30), with final fadeouts.

Short (2:30) and long (7:51) versions of the auditory stimuli in the control condition were similar in duration and intended to be approximately as engaging as listening to the music [27,33]. They comprised a short description of the invention of the television, adapted from a standardized test of listening comprehension [53].

### 2.3. Measures of Cognition and Emotion

The cognitive tasks measured visuospatial WM and two executive functions: (1) verbal fluency and (2) cognitive flexibility and speed. An additional test measured arithmetical ability. Two versions of each task were created to be of equal difficulty so that they could be counterbalanced with pre- and post-testing.

As older adults might be less comfortable with computerized tests, and because both paper-and-pencil and computerized tests are good at assessing cognitive function in older adults [54], we decided to present the tests in paper-and-pencil versions to put participants at ease. Moreover, the actual tasks were selected so that testing would be complete within 15 min after listening [44,45].

Visuospatial WM was tested with the backward Corsi blocks task [55]. Participants were presented with nine blocks arranged randomly on a wooden tablet. The experimenter tapped a sequence of blocks and asked the participant to tap the same blocks in reverse order. Tap sequences increased in length (from 2 to 7), with one trial for each sequence length. The dependent variable was the longest sequence completed successfully (maximum: 6).

Our measure of verbal fluency was adapted from Novelli et al. [56]. Participants were given 1 min to generate as many words as possible that start with a given letter (F or P), excluding proper names. The dependent variable was the total number of appropriate words produced orally by the participant.

Cognitive flexibility and speed were tested with Trail Making Test B (TMT-B) [57]. Here, we administered only TMT-B without TMT-A, as is typically done, to focus on whether the task would be completed more quickly as a consequence of listening to music. Participants were presented with an A4 sheet of paper with 25 circles (2 cm diameter) containing the numbers from 1 to 13 or 12 letters from A to N, all in the same font (Arial 24; J and K from the original test—which are not in the Italian alphabet—were replaced with M and N). They used a pencil to connect the circles in sequential order, alternating between numbers and letters (1-A-2-B-3-C and so on) as accurately and rapidly as possible. The time taken to complete the task (measured in seconds) was the dependent variable.

For arithmetic ability, we used the AC-FL [58], which measures the speed and accuracy with which participants mentally add, subtract, and multiply (blocked presentation). We administered only the addition and subtraction blocks, in that order. For both, participants had 1 min to complete as many operations as possible. The dependent variable was the total number of correct responses.

Finally, we measured emotional state with the Self-Assessment Manikin (SAM) [35], a questionnaire that comprises pictures taken from the International Affective Picture System [59]. In each trial, participants viewed a single picture and rated their felt emotional response using three nine-point scales (arousal, mood/valence, and dominance). For each scale, 1 indicated the lowest rating (e.g., low arousal) and 9 indicated the highest (e.g., high arousal).

### 2.4. Procedure

Participants were tested in a single 90-min session, illustrated in Figure 1. After the pre-screening tests (SVAMA, WAIS-IV Vocabulary, PSW-B, PANAS, and AFML, in that order), they completed the SAM, followed by the cognitive tasks in the following order: TMT-B, backward Corsi blocks, AC-FL, and verbal fluency. After completing the cognitive tasks, participants were assigned randomly to one of six listening conditions formed in a 3 × 2 factorial design based on listening condition (Mozart, Albinoni, control) and duration (short, long). They wore headphones (Sennheiser HD 280 Pro, Wedemark, Germany) while the experimenter watched and ensured that they were not distracted. After the listening session, participants completed the SAM again, followed by alternative versions of the cognitive tasks in the same order.

## 3. Results

To confirm that participants assigned to the six different listening conditions did not differ at pre-test (i.e., before they listened to any music), we used one-way between-subject analyses of variance (ANOVAs) to test for differences in demographics (age, education) and the pre-screening measures (SVAMA, Vocabulary, PWB-Q, PANAS-Positive, PANAS-Negative, AFML). There were no group differences (*p*s ≥ 0.08). Descriptive and inferential statistics are provided in the Appendix A. We also confirmed that there were no gender differences in the cognitive or emotion variables at pre- or post-test (*p*s ≥ 0.4, Bonferroni-corrected for seven tests) and no correlations with age (*p*s ≥ 0.09). Gender and age were not considered further.

The analyses that follow consist of: (1) group comparisons for the cognitive tasks, (2) group comparisons for the emotion measures, and (3) associations between cognition and emotion scores; specifically, those that showed reliable group differences in the previous analyses.

### 3.1. Cognitive Tasks

Preliminary analyses confirmed that there were no group differences at pre-test for the four cognitive tasks (*p*s > 0.6). Statistics are provided in the Appendix A. Mixed-design ANOVAs with one repeated measure (pre- or post-test) and two between-subject factors (listening condition, duration) were conducted separately for each of the four cognitive variables. Means and *SD*s are provided in Table 1. To control for the possibility of inflated type I error because we had four dependent variables, we Bonferroni-corrected the alpha level for the crucial test of the interaction between test session and listening (α = 0.0125).

For WM, there was a main effect for test session, with improvement from pre- to post-test (*F*(1, 126) = 11.25, *p* < 0.001, partial η^2^ = 0.082). This main effect was qualified by a two-way interaction between test session and listening condition (*F*(2, 126) = 7.24, *p* = 0.001, partial η^2^ = 0.103) (Figure 2). There were no other main effects or interactions (*p*s > 0.2). Follow-up tests revealed that, although there was large pre-to-post improvement in WM for the Mozart group (*p* < 0.001), there was no change for the Albinoni (*p* = 0.263) or control (*p* = 0.508) groups.

Analysis of verbal-fluency scores revealed an improvement from pre- to post-test (*F*(1, 126) = 8.36, *p* = 0.005, partial η^2^ = 0.062) but no other main effects or interactions (*p*s > 0.5). The results were identical for cognitive flexibility and speed, with a large improvement (i.e., faster performance) from pre-to post-test (*F*(1, 126) = 140.63, *p* < 0.001, partial η^2^ = 0.527) but no other main effects or interactions (*p*s > 0.1), and for arithmetic, with a substantial improvement over time (*F*(1, 126) = 42.56, *p* < 0.001, partial η^2^ = 0.252), but null otherwise (*p*s > 0.2).

As differential improvement in cognitive performance was evident for only one of four cognitive tasks (WM), we also used Bayesian analyses, conducted with JASP 0.16.3 (default priors) [60], to determine whether the observed WM data were more likely under one of two models: one that included the three main effects (test session, listening condition, duration) and a second that included the main effects plus the interaction between test session and listening condition. The observed data were 31.5 times more likely under the model that included the interaction, which provided very strong evidence for the interaction effect [61]. There was also strong and very strong evidence, respectively, for increases in WM for listeners in the Mozart-short (BF_10_ = 17.8) and Mozart-long (BF_10_ = 32.7) conditions. For the other four groups, the data provided substantial evidence for the null hypothesis (i.e., no changes in WM, all BF_10_ < 0.250).

### 3.2. Emotion Tasks

We used the same analytic method for emotional responses collected at pre- and post-test with the SAM. Means and *SD*s are provided in Table 1. For arousal scores, a marginal effect of music (*F*(1, 126) = 3.01, *p* = 0.053, partial η^2^ = 0.046) was qualified by a two-way interaction between listening condition and test session (*F*(2, 126) = 15.37, *p* < 0.001, partial η^2^ = 0.196) (Figure 3). Arousal levels increased after listening to Mozart (*p* = 0.009) but decreased after listening to Albinoni (*p* < 0.001). For participants in the control condition, they did not change (*p* > 0.5), and there were no other main effects or interactions (*p*s > 0.07).

For mood scores, a significant main effect of listening condition (*F*(1, 126) = 11.09, *p* < 0.001, partial η^2^ = 0.150) was qualified by a two-way interaction between listening condition and test session (*F*(2, 126) = 7.86, *p* < 0.001, partial η^2^ = 0.111) (Figure 4). Follow-up tests revealed significant improvement in mood for the Mozart group (*p* < 0.001) but no improvement for the Albinoni group (*p* > 0.9) and a marginal decrease for the control group (*p* = 0.057). All other tests were non-significant (*p*s > 0.09), except for a two-way interaction between duration and test session (*F*(1, 126) = 7.15, *p* = 0.008, partial η^2^ = 0.054). Improvements in positive mood were evident for participants in the long-duration conditions (*p* = 0.002) but not for those in the short-duration conditions (*p* = 0.444). The three-way interaction was only marginal (*F*(2, 126) = 2.43, *p* = 0.092, partial η^2^ = 0.037), which indicates that two-way interaction did not differ significantly across the three listening conditions.

As stimulus duration is an important methodological issue for research, we opted to examine further effects of duration on mood scores separately for the three listening conditions. For the Mozart group, a two-way interaction between test session and duration (*F*(1, 42) = 11.27, *p* = 0.002, partial η^2^ = 0.212) stemmed from a marked improvement in mood for listeners who heard the long-duration Mozart piece (*F*(1, 21) = 30.72, *p* < 0.001, partial η^2^ = 0.594), but there was no improvement for listeners who heard the short-duration piece (*F* < 1). For the Albinoni listeners, there was no two-way interaction (*p* = 0.308), and no main effects of test session or duration (*p*s > 0.6). For the control group, there was no two-way interaction (*p* = 0.860), no main effect of duration (*p* = 0.590), and a marginal negative effect on mood from pre- to post-test (*F*(1, 42) = 3.82, *p* = 0.057, partial η^2^ = 0.083). Follow-up Bayesian analyses showed that the observed data provided decisive evidence for improvements in mood for listeners in the Mozart-long condition (BF_10_ > 100). For the Mozart-short group, the data actually provided substantial evidence for the null hypothesis (BF_10_ = 0.255). For the Albinoni-short group, small improvements in mood scores were more consistent with the null than the alternative hypothesis (BF_10_ = 0.267). For the other three groups, mood scores actually *decreased* from pre- to post-test, at least in absolute terms.

For dominance scores, there were no main effects or interactions (*p*s > 0.3), as predicted.

### 3.3. Associations between Cognition and Emotion

In the final set of analyses, we asked whether improvements in WM were associated with changes in arousal and mood, calculated as difference scores (post–pre). Increases in arousal were accompanied by improvements in mood (*r* = 0.298, *N* = 132, *p* < 0.001). Changes in WM were *not* associated with increases in arousal (*p* > 0.3) (Figure 5), however, or with improvements in mood (*p* > 0.2) (Figure 6). The lack of a significant association between changes in WM and changes in arousal or mood precluded the possibility of mediation analysis. Moreover, when we used a general linear model to predict improvements in WM as a function of three predictor variables (listening condition, increases in arousal, improvements in mood), the effect of listening condition remained significant (*F*(2, 127) = 6.35, *p* = 0.002, partial η^2^ = 0.091), with larger improvements in WM for the Mozart group than for the Albinoni group (*p* = 0.011) and the control group (*p* = 0.004) but no difference between the Albinoni and control group (*p* > 0.9) (Tukey’s tests). Increases in arousal (*p* > 0.8) and improvements in mood (*p* > 0.9) had no partial association with changes in WM when group differences were held constant.

The null associations between improvements in WM and changes in arousal and mood confirm that the observed data were not particularly unlikely if the true association were null. To explore this null result in more detail, we again used Bayesian analyses to test whether the observed data were more likely under the null (no association) or alternative (association) hypothesis. For the association between changes in WM and increases in arousal, the observed data were 5.59 more likely under the null than the alternative hypothesis. For the association between changes in WM and improvements in mood, the observed data were 4.81 times more likely under the null hypothesis. In other words, the data provided substantial evidence for the null hypothesis in both instances [61]. In contrast, for the association between increases in arousal and improvements in mood, the observed data were 42.5 times more likely under the alternative hypothesis. Even when the Mozart listeners were examined on their own (*n* = 44), there was no association between improvements in WM and changes in arousal (*p* > 0.7) or mood (*p* > 0.2).

## 4. Discussion

We examined whether happy-sounding music improved the emotional state of active, autonomous, community-dwelling older adults and whether such improvement was accompanied by improvement in cognitive performance. As expected, in contrast to listening to sad-sounding music (Albinoni) or to a spoken-word recording, listening to happy-sounding music (Mozart) increased arousal levels and improved mood. In contrast, listening to sad-sounding music was accompanied by reduced levels of arousal. As predicted, there was no effect of music listening on dominance levels. Participants who heard happy-sounding music also exhibited improvements in visuospatial WM. At the group level, then, the results are consistent with the arousal and mood hypothesis. At the individual level, however, improvements in arousal and/or mood had no association with improvements in WM. In fact, the observed data provided substantial evidence for *no* association.

How can we explain these seemingly contradictory results? For younger adults, cognitive benefits emerge reliably after listening to happy-sounding music, and these benefits appear to be a consequence of changes in arousal and/or mood [25,29,30]. We propose that, in some instances, greater individual variability among older compared to younger adults makes it relatively difficult to observe correlations between scores on one task and scores on another task. In fact, previous mood-induction research noted specifically that it is important to consider age-related factors [62]. Emotional responses to music, in particular, show age-related changes, such that older adults have stronger emotional reactions to happy-sounding music compared to music that conveys other emotions [63]. They also have difficulty recognizing negative emotions in music, as well as in speech, although the ability to recognise positive emotions in both domains remains relatively stable throughout adulthood [21,63]. In some cases, however, older adults exhibit recognition deficits for positive and for negative emotions—for music as well for faces [64]—although such deficits appear to be larger for music that conveys negative emotions [65]. The “positivity effect” refers to findings showing that, in general, older individuals preferably attend to and remember positive compared to negative stimuli [66]. These factors could explain why older adults in our study did not show a decrease in mood after listening to the sad-sounding music, even though they had lower arousal levels.

Studies and meta-analyses of reaction-time data also reveal increased intra-individual differences in typical [67] as well as pathological aging [68]. Such trial-by-trial fluctuations in responding for individual older adults make their response times inherently noisy. The lack of an association between emotional response and cognitive ability could also stem from differences in measurement approaches, with our WM task requiring more effort than self-reports of emotional responding [69]. Nevertheless, an association between other self-report emotion tasks and objective cognitive measures has been observed previously with younger adults [25,29,30]. In other words, if effort plays a role, its effect appears to be moderated by age.

Another possibility is that mood induction has an atypical association with cognitive performance in older adulthood. For example, when film clips are used to induce positive or negative moods, performance in a subsequent test of executive function (Tower of London) is impaired for older adults in both mood-induction conditions compared to a neutral control condition [70]. In contrast, younger adults’ performance is similar across conditions. These findings could explain why, in our study, performance did not improve in the two tests of executive functions, even though it did in the test of WM. Tower of London is typically considered to measure planning, and planning is obviously required for good performance in the Trail-Making Test-B. It is difficult, however, to see how planning extends to the verbal-fluency task. There are also many different tests of executive functions, and it seems unlikely that mood induction would have similar detrimental effects for all of them in samples of older adults.

Among adults who vary widely in age, WM is known to be correlated highly with many aspects of executive functioning [71]. In fact, differential performance in our WM task, but not in the other cognitive tasks, might have been because only the WM test was not timed. In contrast, scores in the test of cognitive flexibility and speed (TMT-B) were calculated as the time taken to complete the task. Scores in the verbal-fluency and arithmetic tests—number correct in 1 min—also relied on speed in responding. In general, we know that processing speed starts to slow down after age 20—due to caution in decision making, for example—but it becomes even slower over the age of 60 due to actual slowing of mental speed [72]. The WM task may also have been more difficult in terms of cognitive requirement and attentional control compared to the other tasks, thereby allowing for more room for improvement. In any event, task-specificity remains an unresolved issue for performance after mood induction, in general, and for the so-called Mozart effect, in particular [17].

We also tested whether consequences of music listening would change depending on the duration of music stimuli. Compelling evidence would have come from a significant three-way interaction, indicating that changes from pre- to post-test were evident only for listeners who heard the long-duration Mozart excerpt. This interaction was not significant. Nevertheless, Bayesian analyses showed positive evidence for improvements in WM for listeners in the Mozart group: strong evidence for the short-duration group and even stronger evidence for those who heard the longer duration, although improvements in mood were evident only for listeners in the Mozart-long condition. Clearly, these findings need to be replicated in future research, but the present results do not allow us to recommend using shorter-duration excerpts for applied or research purposes. Indeed, as older people generally need more time to process information, including auditory stimuli [45], they are likely to benefit from longer exposure to music, to experience its full effects on emotion and cognition.

Although pre- and post-testing represented a repeated measure, the listening manipulation was between-subjects, which did not allow us to compare the effect of listening to the different excerpts among the same individuals. In addition to treating the listening manipulation as a repeated measure, future studies could investigate whether repeated, regular sessions of music listening have a cumulative effect on cognitive performance. Our exclusive use of self-report measures for emotional response could also be complemented with psychophysiological measures (e.g., skin conductance), which would allow for a more fine-grained examination of the effect of music listening on older adults.

To conclude, our results contribute to the growing literature on the effect of listening to music on older adults’ emotional states and cognitive abilities. Compared to other participants, older adults who heard happy-sounding music exhibited larger positive changes in arousal, mood, and WM, a finding predicted by the arousal and mood hypothesis [25]. Nevertheless, the independence between measures of emotion and cognition appears to stem from large inter- and intra-individual variability that characterizes the aging process. Thus, for at least some older people, music listening could represent a cost-free and effective means to induce positive emotional states and to enhance and sustain cognitive performance.

## Figures and Tables

**Figure 1 brainsci-12-01567-f001:**
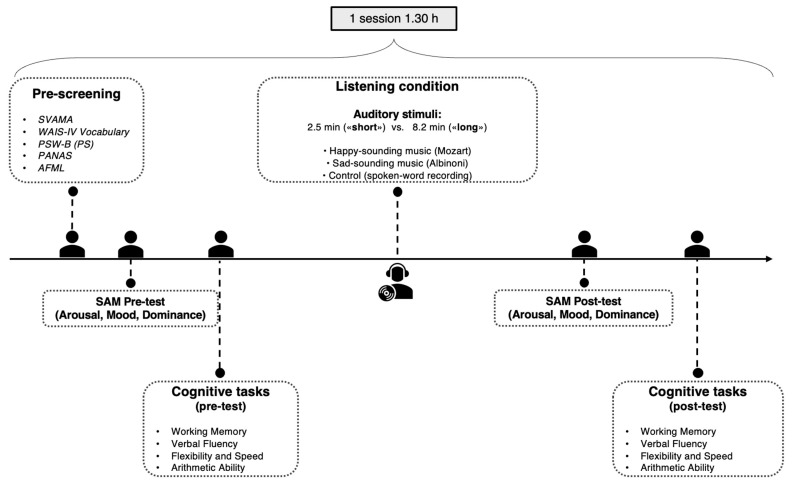
Schematic illustration of the procedure.

**Figure 2 brainsci-12-01567-f002:**
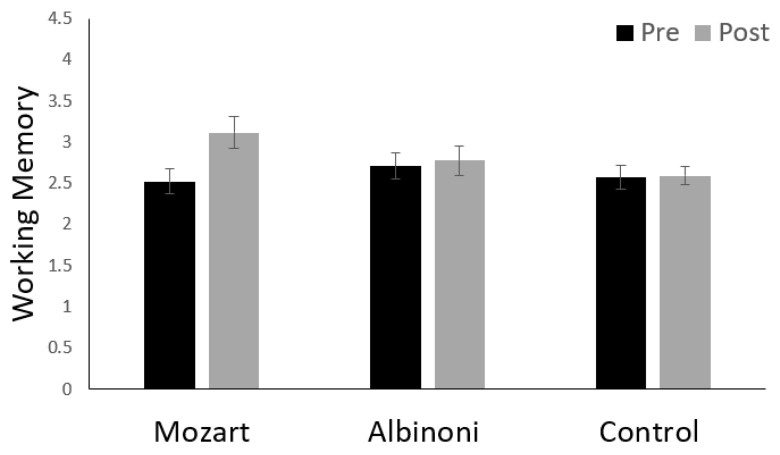
Means at pre- and post-test for scores for the test of working memory (WM). Higher scores indicate better WM. Error bars are *SE*s. The figure illustrates the two-way interaction between test session and listening condition: pre- to post-test improvements were evident only for the Mozart group.

**Figure 3 brainsci-12-01567-f003:**
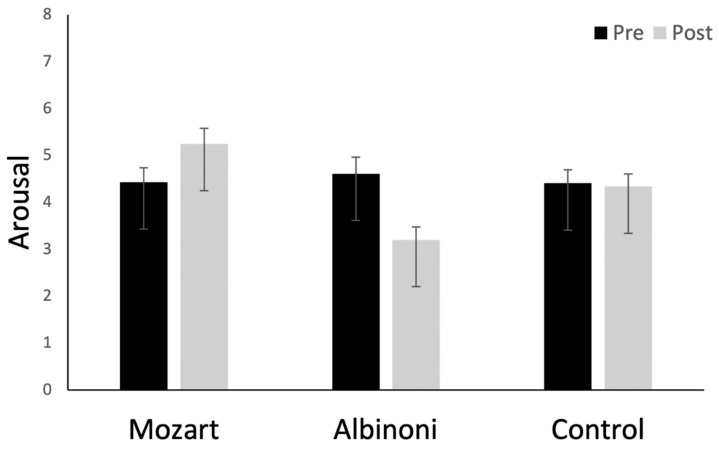
Means at pre- and post-test for arousal scores. Higher scores correspond to higher levels of arousal. Error bars are *SE*s. The figure illustrates the two-way interaction between test session and listening condition: arousal increased in the Mozart condition, decreased in the Albinoni condition, and remained unchanged in the control condition.

**Figure 4 brainsci-12-01567-f004:**
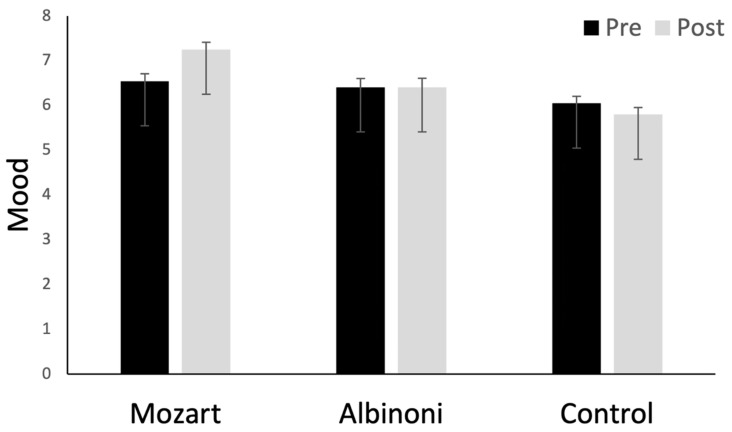
Means at pre- and post-test for mood scores. Higher scores correspond to more positive moods. Error bars are *SE*s. The figure illustrates the two-way interaction between test session and listening condition: mood improved from pre- to post-test in the Mozart condition but was unchanged in the Albinoni condition and declined slightly (but not significantly) in the control condition.

**Figure 5 brainsci-12-01567-f005:**
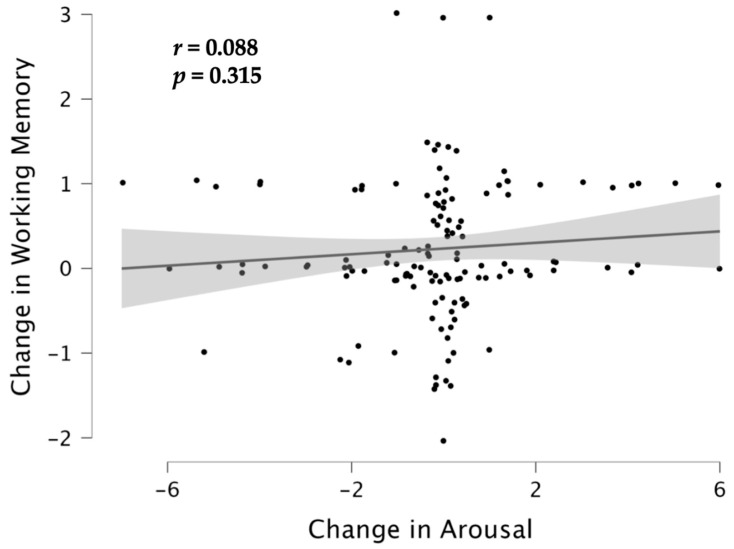
Scatterplot illustrating the null association between change in arousal and change in working memory. The shaded area represents the standard error of the estimate.

**Figure 6 brainsci-12-01567-f006:**
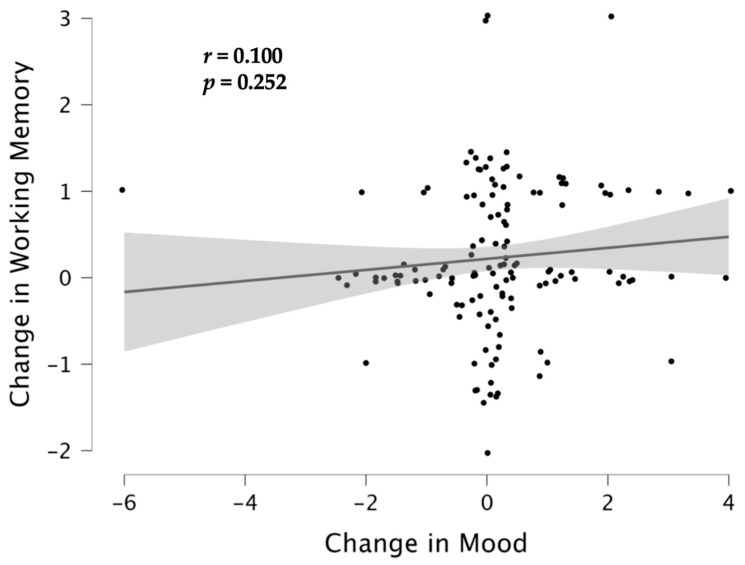
Scatterplot illustrating the null associations between change in mood and change in working memory. The shaded area represents the standard error of the estimate.

**Table 1 brainsci-12-01567-t001:** Descriptive statistics (*M* and *SD*) for measures of cognition and emotion at pre- and post-test ^1^.

		Short Version						Long Version					
		Mozart		Albinoni		Control		Mozart		Albinoni		Control	
		*M*	*SD*	*M*	*SD*	*M*	*SD*	*M*	*SD*	*M*	*SD*	*M*	*SD*
**Cognition**													
Working Memory	Pre	2.73	1.24	2.59	1.10	2.50	0.96	2.32	0.57	2.82	1.01	2.64	1.00
	Post	3.32	1.52	2.68	1.49	2.50	0.67	2.91	0.91	2.86	0.89	2.68	0.78
Flexibility/Speed	Pre	118.50	20.11	117.96	30.74	119.82	23.61	122.05	31.60	123.73	32.03	124.09	35.98
	Post	89.59	27.50	95.23	34.95	106.27	19.88	102.18	29.96	104.00	33.23	102.64	31.16
Verbal Fluency	Pre	11.91	4.02	13.05	3.48	12.46	3.85	12.55	3.84	12.68	4.01	12.68	3.84
	Post	12.86	3.63	13.00	2.33	14.05	4.12	13.05	5.12	13.59	3.51	13.86	3.12
Arithmetic	Pre	38.91	10.94	38.59	7.06	39.18	6.32	36.27	11.27	37.86	9.53	36.82	10.51
	Post	41.36	9.95	40.68	5.31	41.64	5.31	39.05	10.25	40.36	7.31	38.64	9.16
**Emotion**													
Arousal	Pre	4.59	1.41	5.05	2.28	4.73	2.05	4.27	2.51	4.18	2.34	4.09	1.72
	Post	5.14	1.91	3.09	1.41	4.59	2.04	5.36	2.48	3.32	2.17	4.09	1.44
Mood	Pre	6.55	0.96	6.46	1.37	6.14	0.83	5.18	1.22	6.36	1.22	5.96	1.25
	Post	6.68	0.89	6.22	1.34	5.86	0.99	7.82	0.96	6.59	1.30	5.73	1.12
Dominance	Pre	6.64	1.40	7.18	1.33	6.91	2.09	7.05	1.68	6.59	1.62	6.91	1.27
	Post	6.59	1.37	6.55	1.47	7.05	1.21	7.14	1.75	6.50	1.57	6.95	1.53

^1^ All *n*s = 22; see text for details of the specific measures.

## Data Availability

Data supporting the reported results can be obtained by contacting M.V. (margherita.vincenzi@studenti.unipd.it).

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
