# Peer review of "Music Listening, Emotion, and Cognition in Older Adults"

_brainsci, 2022, doi:10.3390/brainsci12111567_

Round 1
Reviewer 1 Report (Previous Reviewer 3)
Thanks for taking care of all my comments.
Author Response
Thank you for all your suggestions.
Reviewer 2 Report (New Reviewer)
This study investigated in 132 individuals whether listening to happy or sad music had an influence on visuospatial working memory, cognitive abilities, verbal fluency, mathematical ability as well as mood and arousal. They find that happy music increased working memory scores, as well as arousal and mood. The manuscript is clearly written, though certain sections are a bit lengthy. The statistical analyses seem appropriate, but an uncorrected p-value is used, while many statistical tests are conducted. Overall, I think the findings are not very straight-forward (out of the 4 cognitive tests, only 1 test specifically points towards an effect of happy music versus sad music/control, that test does not interact with the participant's arousal, they do not test this for the other cognitive abilities) and therefore the conclusion could be more shifted towards having inconclusive findings regarding the mood arousal hypothesis instead of focusing mostly on the effect of happy music on working memory.
Abstract
- It would help to early on mention in the abstract the arousal mood hypothesis, that theories suggest that improved cognitive abilities are due to mood and arousal changes and that that is what is tested in the study. Because the sentence 'at the individual level, however, improvements in WM were unrelated to changes in arousal and mood' makes the reader surprised why you would test that when examining cognitive benefits.
- The last 2 sentences in the abstract are almost a repetition of the 2 before. Possibly a more concluding sentence would be better, like 'this implies … that regardless of the pathway being through mood or arousal or not, music seems to influence some cognitive performances in older adults'
Introduction
- The introduction could benefit from a bit of shortening and more cohesive writing. Especially the descriptions of the measures already discussed in the introduction take up quite a bit of space. It could also just be stated 'that length of listening was taken into account as a potential influence'. And there is a lot of repetition on what this study aims to examine.
- I personally think the part in the first paragraph in which the present investigation is presented could be erased, as the reader is not yet introduced to the mood hypothesis. It also seems to indicate that participants are just 'asked' about the effects instead of tested… And is overlaps with 'the present study'.
- Under present study, stimuli were the same as those used previously. Where previously? By the authors, by a different study?
- Could a definition of dominance be added? Why was exactly this measure chosen as a control?
Methods
- Under participants a description of the sample should be given, not yet results on questionnaires… I would move that information about the participants scores during pre-screening to the result section and all questionnaire information under 'measures'.
- Under participants: What is mean age of the sample and SD and gender distribution? What about education as that suddenly comes up in the results and we have no clue about level of education of the participants. There should be some more descriptives on the participants.
- A reference to the fact that Scimone decreased positive mood? And Mozart increases?
Results
- Figure 3 and 4 seems with overlapping confidence intervals not to suggest the Mozart group significantly differed pre- and post? Are these bars surely correct?
- Somewhere a Bonferonni correction is applied (dividing .05 by only 4 tests), but I think the authors should correct for multiple testing and adjust their alpha with the amount of tests that are done or at least use a more stringent alpha of .01, because many more than 4 tests are conducted.
- Is it a post-hoc decision to test whether WM is associated with changes in arousal and mood? Why not also for processing speed or verbal fluency, which also go up, just not only with happy music but also with sad music? I think it's good to justify this decision as it now looks a little bit like 'picking out' only to do a test when it is in line with the hypothesis.
What about age and gender? Were they added as covariates or controlled for, was their a difference for males and females?
Discussion
- With only 44 people in the Mozart condition, could there be a lack of power to test within people an interaction with arousal on working memory because an effect size of .30 seems quite large to me, is that really what was expected? Why was that effect size chosen? Based on earlier findings?
- Why is limitation that it is a between subject design when there are repeated measures?
- The last concluding sentence that music listening can induce positive emotional states to sustain cognitive performances should be toned down.
Author Response
This study investigated in 132 individuals whether listening to happy or sad music had an influence on visuospatial working memory, cognitive abilities, verbal fluency, mathematical ability as well as mood and arousal. They find that happy music increased working memory scores, as well as arousal and mood. The manuscript is clearly written, though certain sections are a bit lengthy. The statistical analyses seem appropriate, but an uncorrected p-value is used, while many statistical tests are conducted.
***In line with the reviewer’s comment, our use of four cognitive dependent variables provided motivation to lower the alpha-level to p = .0125 (.05/4) for the test of the two-way interaction between test session and listening condition. (Main effects were of no theoretical interest.). The null findings (re the other cognitive measures) are discussed extensively in the discussion.
Overall, I think the findings are not very straight-forward (out of the 4 cognitive tests, only 1 test specifically points towards an effect of happy music versus sad music/control, that test does not interact with the participant's arousal, they do not test this for the other cognitive abilities) and therefore the conclusion could be more shifted towards having inconclusive findings regarding the mood arousal hypothesis instead of focusing mostly on the effect of happy music on working memory.
***The findings are partially consistent with the arousal and mood hypothesis, which we make clear in the Abstract and throughout the manuscript. We have stressed this even more in our new revision. Note that without an interaction between test session and listening condition (i.e., the cognitive variables other than WM), there is nothing further to explain.
Abstract
- It would help to early on mention in the abstract the arousal mood hypothesis, that theories suggest that improved cognitive abilities are due to mood and arousal changes and that that is what is tested in the study. Because the sentence 'at the individual level, however, improvements in WM were unrelated to changes in arousal and mood' makes the reader surprised why you would test that when examining cognitive benefits.
***We re-wrote the Abstract in line with the reviewer’s comments.
- The last 2 sentences in the abstract are almost a repetition of the 2 before. Possibly a more concluding sentence would be better, like 'this implies … that regardless of the pathway being through mood or arousal or not, music seems to influence some cognitive performances in older adults'
***We re-wrote the Abstract in line with reviewer’s comments.
Introduction
- The introduction could benefit from a bit of shortening and more cohesive writing. Especially the descriptions of the measures already discussed in the introduction take up quite a bit of space. It could also just be stated 'that length of listening was taken into account as a potential influence'. And there is a lot of repetition on what this study aims to examine.
***We shortened the introduction and eliminated much of the repetition, including a whole paragraph.
- I personally think the part in the first paragraph in which the present investigation is presented could be erased, as the reader is not yet introduced to the mood hypothesis. It also seems to indicate that participants are just 'asked' about the effects instead of tested… And is overlaps with 'the present study'.
**We re-wrote the first paragraph in line with the reviewer’s comments.
- Under present study, stimuli were the same as those used previously. Where previously? By the authors, by a different study?
***We now include citations to clarify that the stimuli were used in previous (different) studies.
- Could a definition of dominance be added? Why was exactly this measure chosen as a control?
***We now include a definition of the proposed third dimension (dominance: the extent to which the sensations and affective reactions evoked by a stimulus are controllable; Imbir & Gołąb, 2016), which was selected because it is included in the SAM, and considered by some theorists to be a third, independent dimension of emotional responding. Any other choice would be ad hoc.
Methods
- Under participants a description of the sample should be given, not yet results on questionnaires… I would move that information about the participants scores during pre-screening to the result section and all questionnaire information under 'measures'.
***Following the reviewer’s comment, we made the suggested change (see 261-265).
- Under participants: What is mean age of the sample and SD and gender distribution? What about education as that suddenly comes up in the results and we have no clue about level of education of the participants. There should be some more descriptives on the participants.
***We now provide information regarding the mean age, the standard deviation, and the gender distribution of the sample, together with information concerning education (157-162).
- A reference to the fact that Scimone decreased positive mood? And Mozart increases?
***Scimone was the conductor of the Adagio. We included the requested references (Husain et al., 2002) (see 189).
Results
- Figure 3 and 4 seems with overlapping confidence intervals not to suggest the Mozart group significantly differed pre- and post? Are these bars surely correct?
***Yes, that is the point. We expanded the figure legends in order to clarify what they are illustrating.
- Somewhere a Bonferonni correction is applied (dividing .05 by only 4 tests), but I think the authors should correct for multiple testing and adjust their alpha with the amount of tests that are done or at least use a more stringent alpha of .01, because many more than 4 tests are conducted.
***This point is moot because the observed p-value for the test of the interaction between test session and listening condition (< .001) would withstand any correction for multiple tests—and this very low p-value was evident for WM, arousal, and mood. Nevertheless, in line with the reviewer’s comment, we Bonferroni-corrected the critical alpha-level for the test of the interaction (266-269).
- Is it a post-hoc decision to test whether WM is associated with changes in arousal and mood? Why not also for processing speed or verbal fluency, which also go up, just not only with happy music but also with sad music? I think it's good to justify this decision as it now looks a little bit like 'picking out' only to do a test when it is in line with the hypothesis.
***The hypothesis applied, a priori, to ALL of the cognitive tests. As noted, without an interaction between test session and listening condition, there was nothing to explain. The main effects (i.e., changes from pre- to post for all conditions, differences between conditions at pre and post) were of no theoretical interest.
What about age and gender? Were they added as covariates or controlled for, was their a difference for males and females?
***We now include information about age and gender at the beginning of the Results section. Because there were no age and/or gender effects, these variables were not treated as covariates.
Discussion
- With only 44 people in the Mozart condition, could there be a lack of power to test within people an interaction with arousal on working memory because an effect size of .30 seems quite large to me, is that really what was expected? Why was that effect size chosen? Based on earlier findings?
***Yes—tradition (e.g., Cohen 1988) holds that this effect size is “medium”.
- Why is limitation that it is a between subject design when there are repeated measures?
***The listening manipulation was between subjects. We no longer describe it as a limitation, but we believe that it would be interesting to test this variable as a within-subjects manipulation.
- The last concluding sentence that music listening can induce positive emotional states to sustain cognitive performances should be toned down.
***Thank you. Done.

Round 2
Reviewer 2 Report (New Reviewer)
I think the manuscript can be accepted in its current form.
This manuscript is a resubmission of an earlier submission. The following is a list of the peer review reports and author responses from that submission.
Round 1
Reviewer 1 Report
The paper presents a well designed study with a large number of participants. The study grounds very much on previous studies with comparable intention being processed with different target groups. It is absolutely reasonable, to shift the focus on the group of elderly people.
The paper outlines the analysis process of the data and reflects on possible reasons for the result. Figure 1 is very good for making the process clear. Figure 2 and 3 illustrate what is described well and easy in the text, here it could be helpful to think about having figures of the cross associations explained in 3.3 for clarification.
As the previous studies have been processed with young adults and children, one of the discussion points is to what extent the results relate to the target group. This is also related to the length of the musical stimuli. Here, I am very happy to see the discussion of the length-aspect. However, I wonder that the musical examples are not questioned at all.
In the description of the music selection, there are some information from which I don't know why they are relevant (sonata form: does this relate to the fact that you choose a part you consider ‚happy‘?). I do appreciate that you looked into the genesis of the pieces, however, I would recommend to reflect also on the kind of music you provide and to what extent this could relate to your results especially concerning mood. Not all music in written in minor is ‚sad‘, not all classical music is perceived without presumptions due to media exposure (where again the age comes in).
Please check the position of the word "however" in the sentences.
Reviewer 2 Report
This manuscript presents findings from an experimental study that examined the effects of happy-or-sad, short-or-long, classical music listening on various measures of cognitive performance, mood, and arousal among older persons in Italy. The control condition was a spoken-word recording listening. The authors reported in the abstract general improvements in all cognitive tasks and under all conditions, with the greatest improvements in working memory (WM), mood, and arousal under the happy-music condition. The improvements in WM were not related to those in affect (meaning mood, arousal, or both?) at the individual level (meaning irrespective of the happy/sad music/control group membership?). But then it is stated that at the group level (meaning when the happy/sad music/control group membership is factored in?), improvements in arousal and mood (meaning affect here as well?) are correlated with cognitive performance (meaning WM or all cognitive measures?). It seems also that there has been a moderation analysis, where the authors examined whether the link between music listening (meaning dividing the participants in this analysis into two groups instead of three, one group that listened to music, and one group that was the control group?) and cognitive performance is mediated by arousal or mood. There was no evidence of such mediations.
I think that this is an empirical study that has a solid base. However, and as the attempt to understand the manuscript’s abstract reveals, I was very confused, and I got lost quickly by reading the methods and discussion sections. Here are only my initial comments on the manuscript:
Introduction: I am not a native speaker in English, but the manuscript seems to be written well. However, terms such as “elderly” should be avoided as they are not considered politically correct. The article seems to depart from a perspective of active and healthy ageing, with little or no problematization on what these concepts represent in the context of the study. The authors claim in line 30 that they examined autonomous older individuals, but I could not review how this independence was assessed. The citations n47 seems to describe the instrument (SVAMA) that (presumably) is used in this study to assess independence, but since it is in Italian, I could not review what this instrument measures, nor the authors provide any relevant information in the rest of the manuscript.
In line 112, the short-or- long periods of time listening to music are introduced as experimental conditions in the study, although not mentioned at all in the abstract. In line 128, the concept of dominance is introduced as a different measure than mood and arousal yet related to affect. At this point I was wondering whether it is dominance what the authors described in the abstract as affect. In line 129, the authors state certain hypotheses, but then in line 142 they state that this study was exploratory. And then comes in line 147 a research question on whether “music listening influences affective states and cognitive performance”, which came as puzzling, since a music/no music condition is stated here, but no happy/sad music/control condition, nor the short-or-long periods of music listening condition. And even more puzzling was that although it was stated that this is an exploratory study, another set of hypotheses are articulated, starting from line 149.
Not wishing to further make the editor and the authors tired, I must state that I do not have any confidence that this manuscript merits publication, even if it will become subjected to major revisions. That is not to say that I withdraw my trust from the study, which as stated, I think that has a solid ground. But the authors should consider proceeding with major revisions and resubmitting this article anew, making concepts and measures clear from the beginning. Particular attention should be places in the analysis section, as I was unable to follow what analysis corresponded to what hypothesis testing. A section that details the analysis plan is strongly recommended.
Reviewer 3 Report
I have several issues with this study, first, I found the intro long and not very structured, thus is did not get why this study is necessary and how it will bring forward the field of cognitive enhancing activities older adults might engage in.
Next I wondered about the statement that cognitive function decline overall, given recent meta analyses do no longer support this broad claim (i.e. Rey-Mermet & Gadem PBR, 2018). Next, why was a paper pencil method and no computerized test was chosen and why only one task per executive function facet. Next, the TMT A measures performance speed and should always be included when using the TMT B. What motivated the length manipulation given it was a one time intervention? Why were the authors engaging in a one time intervention at all? Why was speech chosen as control condition and instrumental music as intervention? Given much work on training, having 20 something or even less participants per group is really outdated and the results, also I appreciate the effort seem random at best.